# Solar Background Radiation Temperature Calibration of a Pure Rotational Raman Lidar

Vasura Jayaweera[1], Robert J. Sica[1], Giovanni Martucci[2], and Alexander Haefele[2,1]

[1]Department of Physics and Astronomy, The University of Western Ontario, London, N6A 3K7, Canada
[2]Federal Office of Meteorology and Climatology, MeteoSwiss, CH-1530 Payerne, Switzerland

**Correspondence:** Alexander Haefele (alexander.haefele@meteoswiss.ch)

**Abstract.** Raman lidars are an important tool for measuring important atmospheric parameters including water vapor content and temperature in the troposphere and stratosphere. These measurements enable climatology studies and trend analyses to be performed. To detect long-term trends it is critical to have as reliable and continuous as possible calibration of the system and monitoring of its associated uncertainties. Here we demonstrate a new methodology to derive calibration coefficients for a rotational Raman temperature lidar. We use solar background measurements taken by the rotational Raman channels of the Raman Lidar for Meteorological Observations (RALMO) located at the Federal Office of Meteorology and Climatology MeteoSwiss in Payerne, Switzerland, to calculate a relative calibration as a function of time, which is made an absolute calibration by requiring only a single external calibration, in our case with an ensemble of radiosonde flights. This approach was verified using an external time series of coincident radiosonde measurements. We employed the calibration technique on historical measurements that used a Licel data acquisition system and established a calibration time series spanning from 2011 to 2015 using both the radiosonde-based external and solar background-based internal methods. Our results show that using the background calibration technique reduces the mean bias of the calibration by an average of 0.5 K across the troposphere compared to using the local radiosoundings. Furthermore, it demonstrates the background calibration's ability to adjust and maintain continuous calibration values even amidst sudden changes in the system, which sporadic external calibration could miss. This approach ensures that climatological averages and trends remain unaffected by the drift effects commonly associated with using daily operational radiosondes. It also allows a lidar not co-located with a routine external source to be continuously calibrated once an initial external calibration is done. Furthermore, the technique works both for temperature retrievals using the optimal estimation method and the traditional temperature algorithms.

## 1 Introduction

Water vapor is the predominant greenhouse gas, with its abundance significantly regulated by surface temperature. When air temperature rises, the Clausius-Clapeyron equation predicts that the equilibrium vapor pressure of water will increase, leading to higher levels of water vapor in the atmosphere. Positive climate feedback, caused by an increase in water vapor concentration, ultimately leads to elevated temperatures (Colman and Soden, 2021; Dessler et al., 2013; Held and Soden, 2000). Accurate retrievals are crucial for conducting precise relative humidity (RH) climatology and trend studies in the Upper Troposphere

and Lower Stratosphere (UTLS) region with Raman lidar measurements. Consequently, the credibility of the computed trends relies significantly on the reduction of uncertainties associated with these measurements. Direct retrieval of RH from Raman lidar measurements necessitates the calibration of temperature measurements, and a notable contributor to the uncertainty budget in Raman lidar measurements stems from the determination of these temperature calibration constants. Enhancing and refining these calibration methods are important steps toward achieving greater accuracy and reducing uncertainties in our investigations. Mahagammulla Gamage et al. (2019) proposed an Optimal Estimation Method (OEM) based methodology for temperature retrieval that considers the full Raman lidar equation, without requiring the assumption of an empirical calibration function. This approach mitigates uncertainties when contrasted with the utilization of empirical calibration functions, which could potentially introduce substantial errors exceeding 1 K, particularly in cases involving larger temperature ranges (Behrendt, 2005). However, it is crucial to recognize that the accuracy of any calibration method utilizing radiosondes depends on the uncertainty associated with the reference radiosondes. Sherlock et al. (1999) proposed an alternative approach known as the background calibration method, for calibrating water vapor mixing ratio measurements obtained through Raman backscatter water-vapor lidar systems. Their method is classified as an internal calibration technique. This method was further expanded by Hicks-Jalali et al. (2018) to generate a time series for water vapor calibration using Raman Lidar for Meteorological Observations (RALMO) data. This method uses the ratio of the solar background signal in detector channels to deduce a calibration constant. In this study, we will adapt this internal calibration technique to produce temperature calibration values for a rotational Raman temperature lidar. This approach distinguishes itself from the external method by enabling the calculation of the complete calibration time series through a single calibration, achieved using an ensemble of external calibrations. The ensemble reduces the systematic uncertainties introduced by the external calibrations, resulting in a more robust calibration time series. Consequently, this allows the establishment of a temperature calibration time series whose temporal evolution is independent of subsequent external measurements. Although an extensive ensemble of radiosondes provides the most robust results, this approach can still be applied with a limited number of radiosondes, but with potentially reduced precision. This ensures that sites with only a few available soundings can still utilize the method effectively. This methodology offers the prospect of generating temperature and RH trends that are free from the influences of radiosonde drifts.

## 2 Measurements and Methodology

### 2.1 Raman Lidar for Meteorological Observations (RALMO)

In order to develop our method, we used Raman lidar measurements obtained from the RALMO. The lidar is located in Payerne, Switzerland at the facility of the Federal Office of Meteorology and Climatology (MeteoSwiss, $46°48'$ N, $6°56'$ E, $492$ m a.s.l) and has been in near-continuous operation since 2009. RALMO was constructed at the École Polytechnique Fédérale de Lausanne (Dinoev et al., 2013). RALMO's configuration includes a narrow field-of-view lidar receiver and a frequency-tripled Nd:YAG Q-switched laser producing an energy output of 300-400 mJ per pulse at 355 nm and at 30 Hz, and is capable of taking measurements continuously during both daytime and nighttime. RALMO's data acquisition is performed using Licel GmbH transient recorders, which enables simultaneous measurement of atmospheric signals through two distinct methods: photon

counting and analog detection. This system utilizes a 250 MHz photon counter in conjunction with a 12-bit, 40 MSPS analog digitizer. The system achieves a minimum time resolution of 25 ns, corresponding to a vertical resolution of 3.75 meters in altitude. In August 2015, RALMO's data acquisition was transitioned from the Licel system to the more advanced and efficient FAST ComTec P7888 (FastCom) data acquisition system (Martucci et al., 2020). Consequently, the dataset is divided between data collected using the Licel system and that acquired with the FastCom system. However, for the purposes of this study, we focus exclusively on the historical data obtained through the Licel acquisition system. RALMO uses a polychromator designed for Pure Rotational Raman (PRR) spectroscopy, allowing it to isolate Rayleigh and Mie lines, including the Cabannes line. PRR spectra from diatomic molecules like $N_2$ and $O_2$ have rotational lines spaced on both sides of the exciting wavelength (Stokes and anti-Stokes branches). Analyzing certain lines or groups of adjacent lines enables the retrieval of vertical temperature profiles in the troposphere and lower stratosphere, as the intensity of these spectra is sensitive to temperature and wavelength (Dinoev et al., 2010; Whiteman, 2003). Various validation studies have been conducted to assess the accuracy of RALMO measurements of temperature and water vapor. Brocard et al. (2013b) conducted a validation study focusing on RALMO measurements of water vapor, employing collocated radiosondes. Their findings indicate that, on average, the water vapor mixing ratio closely matched radiosonde values, with differences of approximately 5 to 10% up to 8 km during nighttime and within 3% up to 3 km during daytime operations. Martucci et al. (2021) compared RALMO measurements with measurements from two reference operational radiosounding systems (ORSs) co-located alongside RALMO. Their findings demonstrate that RALMO measurements meet the OSCAR (Observing Systems Capability Analysis and Review Tool) requirements at breakthrough level for high-resolution numerical weather prediction (NWP) in the free troposphere in terms of measurement uncertainty and observing cycle. (https://space.oscar.wmo.int/requirements, accessed on 3 April 2024).

## 2.2 Radiosondes

Since October 2011, MeteoSwiss has been conducting biweekly launches of Vaisala RS92 radiosondes at 11:00 and 23:00 UTC retrieving atmospheric profiles of temperature, humidity, pressure, and wind. In 2012 MeteoSwiss became a part of the Global Climate Observing System (GCOS) Reference Upper-Air Network (GRUAN) with the Vaisala sonde RS92. As a result, these radiosonde datasets have been reprocessed by collecting metadata, applying correction algorithms, and performing uncertainty estimates, to produce a GRUAN-certified data product (Dirksen et al., 2014). In late 2013, Vaisala introduced the RS41 radiosonde, marking the fourth generation of their atmospheric profiling instruments. This new model was designed to replace the RS92 radiosonde and brought enhanced precision in measuring atmospheric variables. The RS41 radiosonde features advanced sensor technologies, along with cutting-edge design and manufacturing techniques. These improvements, combined with its ease of use, deliver reliable and highly precise atmospheric measurements (Jensen et al., 2016; Dirksen et al., 2020).

The Payerne radiosonde PTU (pressure, temperature, and relative humidity) time series has been the subject of a complete reevaluation that led to the homogenized PTU series up to 2011(Brocard et al., 2013a). A more recent analysis describes the homogenization procedure of the entire PTU time series from 1954 to 2022 (Martucci et al, in preparation). The analysis applies two main corrections to the PTU series with respect to the operational radiosonde of Payerne, the Vaisala RS41: (a)

all soundings during the period from 1980 to 2011 have been corrected for residual systematic biases and (b) for statistically significant and traceable breaks along the period from 1954 to 2022 for the RS41. MeteoSwiss has carried out several intercomparison flights of the different radiosonde models with either the reference RS41 or the transfer radiosonde model, the Vaisala RS92. These intercomparison flights have allowed to determine transfer functions to correct for the systematic biases of previous radiosonde models with respect to the RS41 radiosonde, for 11 UTC and 23 UTC flights. With respect to the previous homogenization, this corrects the temperature and RH data for the effects of solar radiation on the temperature sensor according to the changes in radiosonde models that occurred between 2011 and 2018.

## 2.3 Utilizing the Optimal Estimation Method (OEM) for Retrieving Temperatures through PRR Spectroscopy

Sica and Haefele (2015, 2016) introduced a methodology that uses the OEM to retrieve Rayleigh-scatter temperature and vibrational Raman scatter water vapour mixing ratio. Their methodology has several advantages over the traditional techniques, including a full error budget and the determination of instrument averaging kernels. Recent studies by Hicks-Jalali et al. (2020) and Gamage et al. (2020) have further extended the application of OEM to RALMO retrievals to determine water vapour mixing ratio trends, rotational Raman temperature, and RH. OEM, being an inverse technique, employs Bayesian statistics to estimate a target atmospheric parameter by utilizing both a forward model, which encapsulates the complete physics of the measurement process, and a comprehensive description of the instrumentation employed for data acquisition (Rodgers, 2000). This method can be mathematically represented as follows:

$$\mathbf{y} = \mathbf{F}(\mathbf{x}, \mathbf{b}) + \epsilon \tag{1}$$

where $\mathbf{y}$ is the quantity measured, $\mathbf{F}$ the forward model, $\mathbf{x}$ the state vector, $\mathbf{b}$ the model parameter vector, and $\epsilon$ the experimental error. The model parameter comprises variables that are essential for evaluating the forward model but are not directly retrieved. To ensure the reliability of the retrieval process, the uncertainties associated with these model parameters must be well-characterized and subsequently carried through the retrieval process (Rodgers, 2000). The retrieval process leverages Bayes' theorem, which hinges on conditional probabilities, to derive the desired state vector from the measured data. This theorem relies on assessing the probability of a specific outcome by considering prior knowledge of conditions relevant to that outcome. Therefore, *a priori* estimate ($\mathbf{x_a}$) of the state can be used to obtain a statistical estimate for the state vector. By assuming that the measurement state and *a priori* state are Gaussian, the most likely *a posteriori* state can be found by minimizing the cost function using the vectorized form of Bayes' theorem.

$$\text{cost} = [\mathbf{y} - \mathbf{F}(\hat{\mathbf{x}}, \mathbf{b})]^{\mathbf{T}} \mathbf{S}_{\epsilon}^{-1} [\mathbf{y} - \mathbf{F}(\hat{\mathbf{x}}, \mathbf{b})] + [\hat{\mathbf{x}} - \mathbf{x_a}]^{\mathbf{T}} \mathbf{S_a}^{-1} [\hat{\mathbf{x}} - \mathbf{x_a}] \tag{2}$$

where $\mathbf{S}_{\epsilon}$ is the measurement error covariance and $\mathbf{S_a}$ the *a priori* error covariance. The cost function evaluates how well a solution fits the data, and for effective models, the cost is typically close to one. In our validation process, we have chosen to adopt the methodology introduced by Mahagammulla Gamage et al. (2019) for the retrieval of temperature from PRR lidar measurements. Their OEM uses the full physics of PRR scattering to retrieve profiles of temperature directly from the raw measurements, including a profile-by-profile uncertainty budget.

## 2.4 External Calibration for Temperature

Lidar temperature measurements, including those from Raman lidar studied here, require calibration to derive accurate absolute temperature measurements. Mahagammulla Gamage et al. (2019) obtained RH directly from RALMO measurements, using an external calibration method of temperature that relies on an external reference instrument, like a balloon-borne radiosonde. The Raman lidar equation for the backscattered signal $N_{JX}$, where $X$ denotes either $H$ or $L$ corresponding to the high J or low J rotational Raman channel, is given by,

$$N_{JX,t}(z) = C_{JX} \frac{O(z)}{z^2} n(z) \Gamma_{\text{atm}}^2(z) \left( \sum_{i=\text{O}_2,\text{N}_2} \eta_i \sum_{J_i} \tau_{JX}(J_i) \left( \frac{d\sigma}{d\Omega} \right)_\pi^i (J_i) \right) + B_{JX}(z), \tag{3}$$

where $N_{JX,t}(z)$ is the true backscattered signal as a function of altitude $z$, $C_{JX}$ the lidar constant, $O(z)$ the overlap, $n(z)$ the number density of the air molecules, $\Gamma_{\text{atm}}$ the atmospheric transmission, $\eta_i$ the volume mixing ratio of nitrogen and oxygen, $\tau_{JX}(J_i)$ the transmission of the receiver at the wavelength of the rotational Raman line $J_i$, $\left( \frac{d\sigma}{d\Omega} \right)_\pi^i (J_i)$ the differential backscatter cross-section, and $B_{JX}(z)$ the background signal. Following the methodology of Mahagammulla Gamage et al. (2019) we can define a calibration constant, referred to as ($C^*$), as follows:

$$C^* = \frac{C_{JH}}{C_{JL}}, \tag{4}$$

where $C_{JH}$ and $C_{JL}$ represent the lidar constants for RALMO's high J and low J rotational Raman channels, respectively. In this work, we adopt a slightly modified notation from that used by Mahagammulla Gamage et al. (2019). Specifically, we use $C^*$ instead of $R$ to denote the calibration constant in our equations. This change is intended to maintain consistency with our existing notation and to avoid potential confusion with other variables commonly represented by $R$ in related literature. Combining the Raman lidar equation for the backscattered PRR signal with equation 4, we find

$$C^* = \frac{\frac{N_{JH}-B_{JH}}{N_{JL}-B_{JL}}}{\frac{\sigma_{JH}}{\sigma_{JL}}}, \tag{5}$$

where $N_{JH}$ and $N_{JL}$ are the raw signals for high J and low J rotational Raman channels, $B_{JH}$ and $B_{JL}$ the background photon counts for the high J and low J rotational Raman channels, and $\sigma_{JH}$ and $\sigma_{JL}$ denote the term $\sum_{i=\text{O}_2,\text{N}_2} \sum_{J_i} \tau_{JX}(J_i) \left( \frac{d\sigma}{d\Omega} \right)_\pi^i (J_i)$ for the high J and low J rotational Raman channels. For the external method, GRUAN-certified radiosondes launched at nighttime were used. Equation 5 can in principle be evaluated at any altitude, and we have omitted the range dependence for improved readability. We calculated the calibration constants by averaging over the 5 to 8 km altitude range to reduce the random uncertainty and to avoid regions of the profiles where the signals could be saturated.

## 2.5 Internal Calibration for Temperature: The Solar Background Method

External calibration methods necessitate access to an external reference instrument. Depending on the external instrument's operating schedule, these calibration opportunities can be days or weeks apart. Typically, balloon-borne radiosondes serve as the most commonly employed external reference. Calibration using radiosondes can be influenced by the flight path of the

balloon, which, depending on atmospheric conditions, may experience horizontal drift and enter a different air mass compared to what the lidar instrument samples. Such deviations in radiosonde measurements can substantially impact the precision and reliability of the calibration time series. To improve the precision and expand the applicability of external calibration methods, we adopted a technique that computes the relative calibration time series by determining the temporal evolution of the solar background ratio between the high J and low J digital channels. The approach that we present here mirrors Hicks-Jalali et al. (2018) internal calibration method for water vapor mixing ratio, which utilizes the solar background for tracking changes in the mixing ratio calibration constant over time. What distinguishes the approach here is its reliance on a single calibration based on an ensemble of radiosondes to construct the entire calibration time series. This method significantly reduces the uncertainties typically associated with external reference instruments, and makes the calibration time series independent from calibration changes associated with radiosonde measurements. We now define the relative calibration time series $r_{solar}(t)$ as follows:

$$r_{solar}(t) = \frac{B_{JH}^{solar}(t)}{B_{JL}^{solar}(t)},$$

(6)

where $B_{JH}^{solar}$ and $B_{JL}^{solar}$ are the solar background levels detected by the high J and low J rotational Raman channels, respectively. We can now use $r_{solar}(t)$ to calculate the time series of the calibration constant $C^*$. The function is normalized using an ensemble of external calibrations and solar measurements as follows:

$$C^*(t) = \overline{C^*(t)} \frac{r_{solar}(t)}{\overline{r_{solar}(t)}}.$$

(7)

$\overline{C^*(t)}$ is the average of all external calibration points and $\overline{r_{solar}(t)}$ is the average of all background ratios corresponding to the external points, i.e. the background ratio the following morning at a solar zenith angle of $70°$. For our solar background above $55\,\mathrm{km}$, we used the ratio between the solar background from the total counts over 60 minutes from the high J and low J rotational Raman channels. At these altitudes, in a raw 1-minute profile, the lidar signal will be completely due to background solar radiation and not the photons emitted by the laser. Also, we had to consider both the diurnal and seasonal solar cycles when using this solar background method, therefore we chose to only use the solar background at a time corresponding to the lowest solar zenith angle on the winter solstice, which corresponds to a $70°$ zenith angle. We tested the method across various solar zenith angles ($70°, 60°$, and $50°$) and observed an average variation of 0.2% in the calibration constant between these angles. This variation corresponds to a temperature difference of approximately $0.2\,\mathrm{K}$, suggesting the ratio is weakly dependent on the solar zenith angle.

## 2.6 Extending the Background Calibration Technique for Traditional Temperature Algorithms

In this section, we show, how the background calibration can be applied to the traditional temperature algorithms. Following the methodology outlined by Behrendt (2005), $Q(T)$ is defined as follows:

$$Q(T) = \frac{\sum_{i=O_2,N_2} \eta_i \sum_{J_i} \tau_{JH}(J_i) \left(\frac{d\sigma}{d\Omega}\right)_\pi^i (J_i)}{\sum_{i=O_2,N_2} \eta_i \sum_{J_i} \tau_{JL}(J_i) \left(\frac{d\sigma}{d\Omega}\right)_\pi^i (J_i)}.$$

(8)

By using the lidar equation (Equation 3), Equation 8 can be expressed in terms of the background-corrected signals and the calibration constant $C^*$ (Equation 4) as follows:

$$Q(T) = \frac{N_{JH,t}(z) - B_{JH}}{N_{JL,t}(z) - B_{JL}} \times \frac{C_{JL}}{C_{JH}} = \frac{N_{JH,t}(z) - B_{JH}}{N_{JL,t}(z) - B_{JL}} \times \frac{1}{C^*(t)}. \tag{9}$$

For systems that detect only a single PRR line in each of the two PRR channels, Equation 8 can be simplified so that it takes the form,

$$Q(T) = \exp(a - b/T), \tag{10}$$

where $a$ and $b$ are the two calibration constants (Behrendt, 2005). Note that $a$ and $b$ depend on the spectral characteristics of the receiver. By using Equation 9 in conjunction with Equation 10 the calibration constants $a$ and $b$ can be calculated using an external temperature measurement (radiosonde). The above method can also be applied to systems that measure multiple PRR lines, requiring higher-order calibration functions that involve additional calibration constants. We tested this method on the traditional temperature algorithm and obtained results comparable to those achieved when it was used in conjunction with OEM. However, in this study, we focus exclusively on the application of the method in combination with OEM.

## 3   Results and Discussion

Figure 1 shows a comparison between the time series of the temperature calibration constants, derived through the application of the external calibration method and the solar background method. As an illustration of the external calibration method, the time series was computed for a selected number of dates spanning from the end of 2011 to the end of 2015, during which MeteoSwiss in Payerne, Switzerland, had been launching Vaisala RS92 and RS41 sondes to obtain GRUAN-certified profiles of temperature and humidity. For every 60 minutes of count data profiles, a profile-by-profile filtering method was implemented to identify and eliminate scans exhibiting significant cloud cover. This approach involved assessing the signal-to-noise ratio (SNR) of the Nitrogen (N2) digital channel, focusing on the average SNR within the 12 to 14 km range. Profiles with an SNR below 1 were discarded. Furthermore, the calibration dataset used dates where the retained profiles, following the cloud-based filtering mechanism, constituted more than 75% of the initial number of profiles. The calibration time series was calculated through the utilization of reference radiosondes launched at nighttime. Additionally, any calibration points exhibiting an uncertainty greater than 5% were excluded from the time series. The background method calibration was performed daily using the procedure discussed above. We then applied the calibration technique to the measurements collected in the last 4 years of RALMO's operation using a Licel acquisition system. One of the prominent features of the calibration time series is the pronounced decline in the calibration constant's value seen from March to May 2012. This change is attributed to an intervention on the system hardware. However, the only detail that the logbook reveals is the replacement of the coaxial cable connecting the low-J channel photomultiplier to the acquisition system. We can see an increased sensitivity in the low J channel following this intervention, which explains the drop in the calibration constant. We can see that the notable drop observed in the external calibration time series is likewise seen in the background calibration time series, thus emphasizing the

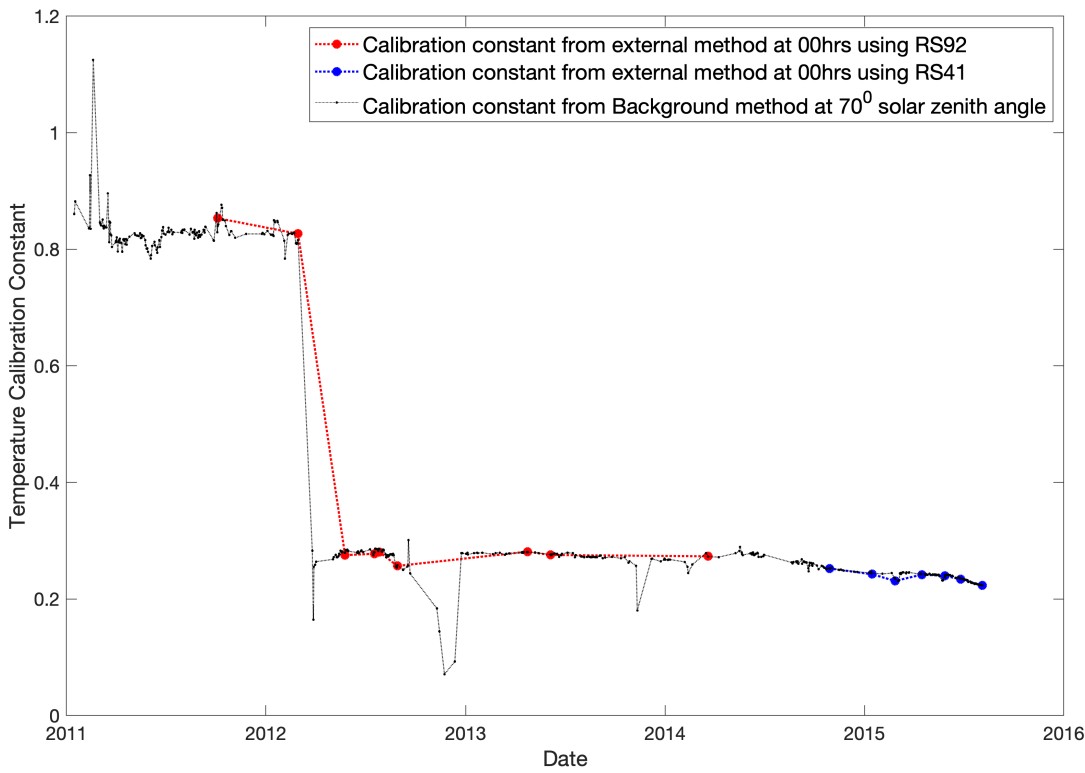

**Figure 1.** Comparison between the temperature calibration constant (dimensionless) obtained by the external method and the temperature calibration constant obtained using the background method. For the external method, the calibration constants were obtained using GRUAN-certified profiles of temperature from Vaisala RS92 and RS41 radiosondes launched at nighttime. For the background method, a solar background above $55\,\text{km}$ from the high J and low J quantum number channels of RALMO at a time corresponding to a $70°$ solar zenith angle was employed.

sensitivity of the background calibration method to changes within the lidar system. This observation highlights the method's ability to measure changes in the system that could be missed with sporadic external calibrations. Also, we can see that the
calibration constant is less noisy after the intervention on the low J channel in 2013. The agreement between the external and background methods is better than 5%. Temperatures were retrieved from the lidar measurements using the OEM-based algorithm presented by Mahagammulla Gamage et al. (2019). Only photon counting measurements were used for the retrievals, as the analog measurements introduced biases that we were not able to correct or explain. The OEM temperature retrieval uses the full physics of PRR scattering and can be calibrated with equation 4 instead of an empirical calibration function.
Additionally, OEMs produce a full uncertainty budget on a profile-by-profile basis while being computationally efficient. The measurements for the years 2011 to 2014 were processed first using the externally determined calibration constants and

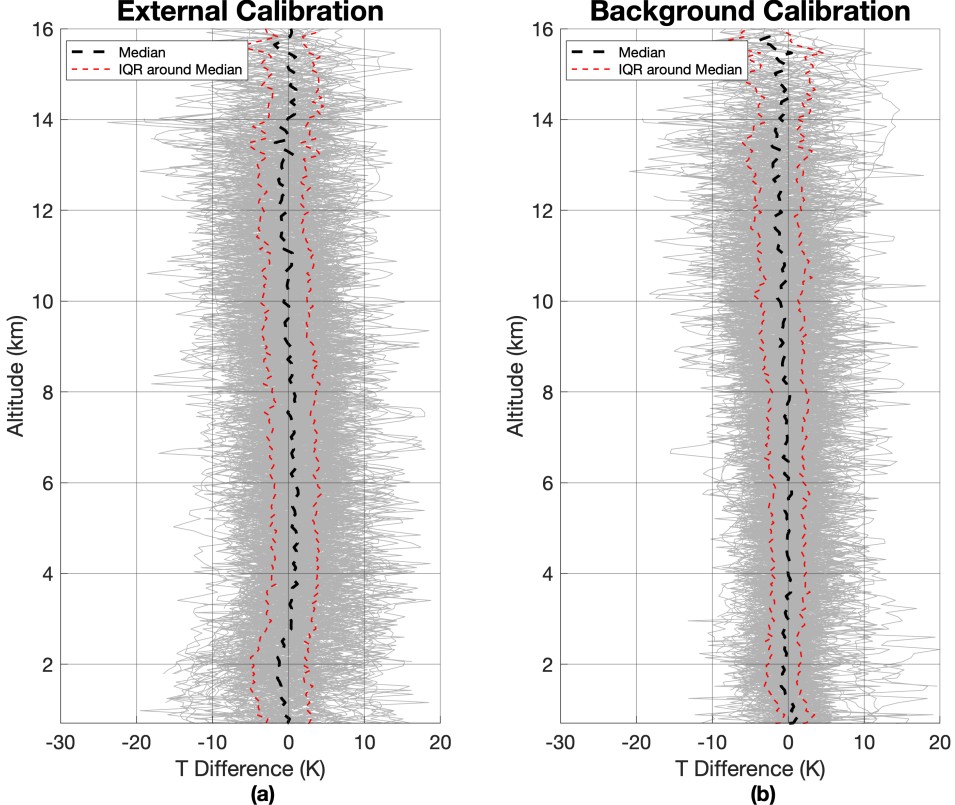

**Figure 2.** (a) The temperature difference between 175 OEM retrieved temperature profiles utilizing external GRUAN-sonde calibration and the homogenized radiosonde temperature profile for the years 2011 (Oct) to 2014 (Dec). (b) The temperature differential observed between 175 OEM retrieved temperature profiles utilizing the solar background calibration and the homogenized radiosonde temperature profile using measurements obtained between October 2011 and December 2014.

secondly utilizing the background method. The externally determined calibration coefficients were interpolated to align with the internal calibration points, resulting in two datasets with identical processed dates for both calibration methods. A filtering method, reliant on the cost associated with the OEM retrieval process was implemented to eliminate bad retrievals from externally calibrated and solar background calibrated datasets (Mahagammulla Gamage et al., 2019). Profiles with a retrieval cost lower than 0.5 or higher than 10 were discarded, indicating overfitting and underfitting, respectively. Furthermore, profiles exhibiting unphysical characteristics in the raw signal were filtered out. These accounted for less than 3% of the total profiles. Each dataset consisted of a total of 175 nights. We also used an upper-cutoff height which was determined as the altitude at which the measurement response function (The area of the temperature averaging kernels) falls below 0.8. Below this specified altitude, the retrieval process is predominantly influenced by the measurements themselves rather than the *a priori* temperature profile. Next, we compared the 175 temperature profiles generated using the two calibration methods with those

**Table 1.** Summary of the mean bias and mean IQR values across different altitude ranges for the temperature difference plots obtained using the external and the background calibration method.

| Calibration Method | Mean Bias (K) | | | | Mean IQR (K) | |
|---|---|---|---|---|---|---|
| | 1-4 km | 4-8 km | 8-12 km | 12-16 km | 1-8 km | 8-16 km |
| External Method | $-0.3 \pm 0.8$ | $0.7 \pm 0.3$ | $-0.2 \pm 0.5$ | $-0.2 \pm 0.8$ | $6.1 \pm 0.6$ | $6.1 \pm 0.7$ |
| Solar Background Method | $-0.2 \pm 0.4$ | $-0.08 \pm 0.2$ | $-0.9 \pm 0.4$ | $-1.4 \pm 0.9$ | $4.3 \pm 0.5$ | $6.0 \pm 1.1$ |

from homogenized radiosonde measurements. Note that the GRUAN-certified radiosondes used for calibration are independent from the homogenized radiosonde data set used for validation. Figure 2a shows the temperature differences between the OEM-derived profiles utilizing the external method and corresponding temperature profiles from the homogenized radiosonde dataset, while Figure 2b shows the comparison with the background calibration method. Table 1 summarizes the mean bias and mean Inter Quartile Range (IQR) values for the two distinct calibration methods across various altitude ranges, corresponding to the temperature difference comparison plots.

For the externally calibrated temperatures (Figure 2a) between 1 to 4 km, a negative mean bias of $-0.3\,\mathrm{K}$ is observed, indicating an underestimation of the lidar-derived temperatures with respect to the radiosonde measured temperatures. This negative mean bias predominantly originates from temperature retrievals obtained between February and October 2012. This period coincides with the large decline in the calibration constant time series, attributed to changes made to the RALMO system. For the subsequent altitude range of 4 to 8 km, a positive mean bias of $0.7\,\mathrm{K}$ is observed, suggesting an overestimation in the lidar-derived temperatures within this interval. For the altitudes, 8 to 12 km and 12 to 16 km an underestimation of temperature values is observed with a negative mean bias of $-0.2\,\mathrm{K}$.

The comparison between the solar background and external calibration methods indicates that for four of the six metrics presented— the mean bias in the 1–4 km, 8–12 km, and 12–16 km ranges, as well as the mean IQR in the 8–16 km range are comparable within the 1-sigma uncertainty level. However, differences are observed in the mean bias for the 4–8 km range and the mean IQR for the 1–8 km range. Specifically, the IQR values for the external method are $6.1\,\mathrm{K}$ for both the 1–8 km and 8–16 km ranges, whereas the corresponding values for the background method are $4.3\,\mathrm{K}$ and $6.0\,\mathrm{K}$, respectively. These observations suggest that the two calibration methods generally yield similar results across most metrics, with the background method demonstrating reduced variability in certain cases.

## 4   Conclusions

We have shown the solar background calibration method is a viable method for the temperature calibrations of rotational-Raman lidars. By using the solar background values acquired by the lidar, this technique provides a more extensive and continuous calibration timeline, which can decrease the difference between the lidar and radiosonde temperatures. Notably, our study highlights the method's adaptability, showcased through its ability to swiftly adjust to modifications within the RALMO system and

demonstrate its responsiveness to system variations that sporadic external calibrations could miss. Our study highlights that the solar background value is weakly dependent on the solar zenith angle, underscoring the robustness of the technique. This potentially enables broader applicability and might simplify implementation under diverse observational conditions, empha-

260 sizing its potential for reliability and widespread use. Moreover, the solar background calibration method offers the advantage of generating a daily calibration timeline based on a single or ensemble of external reference instrument measurements, which mitigates the impacts of drifts and other possible interpretation problems with comparisons to radiosondes. The solar background method is applicable to any PRR temperature lidar and can be used for temperature retrievals using both the OEM and traditional temperature algorithms. The adoption of the background calibration method presents substantial benefits, especially

for climatology and trend studies within the troposphere and lower stratosphere. Its application ensures that climatological assessments and trend derivations remain independent of drift effects associated with radiosonde measurements.

*Data availability.* Measurements used in this paper may be requested from MeteoSwiss by contacting Alexander Haefele (alexander.haefele@meteoswiss.ch).

*Author contributions.* VJ was responsible for the development of the background calibration method code, generating the temperature cali-

270 bration time series. VJ was also responsible for processing the RALMO dataset and carrying out the validation of the calibration constants using radiosonde temperature measurements. He also wrote the initial draft of the paper. RJS wrote the underlying code for the temperature retrieval. RJS and AH defined the project scope and contributed to manuscript preparation. GM helped to validate the calibration time series by providing access to the RALMO and radiosonde measurements and provided expertise on RALMO. Furthermore, GM helped validate the extension of the solar background method for traditional temperature algorithms.

*Competing interests.* The authors declare that they have no conflict of interest.

*Acknowledgements.* We thank Dave Whiteman for his valuable suggestions and for encouraging us to extend the method to traditional temperature determination. We also appreciate the insightful comments and improvements suggested by the anonymous reviewer. We further thank Ludovic Renaud and the entire MeteoSwiss team who operate the Raman lidar. This project has been funded in part by the National Science and Engineering Research Council of Canada and the CASSAVA PEARL (Canadian Anchor Sites for SATellite Validation - Polar

Environment Atmospheric Research Laboratory) project, supported by the Canadian Space Agency, grant number 19FATORA07.

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
