# Peer review of "Solar Background Radiation Temperature Calibration of a Pure Rotational Raman Lidar"

_EGUsphere, 2024_

## Referee Comment (RC1)

**Solar Background Radiation Temperature Calibration of a Pure Rotational Raman Lidar, Jayaweera et al. – a Review**

David N. Whiteman, Howard University

General Comments: This is clearly a good idea and the authors are commended on being the first to demonstrate this new calibration idea which many may react to by saying "why didn't I think of that?". There will no doubt be some iteration on how to best implement the use of background signal in the two rotational Raman channels for temperature calibration but I suspect that eventually this will become the preferred method of calibrating a RR temperature lidar. The authors' results certainly point in that direction although I do have some comments and questions that I hope might help to improve the technique and solidify this publication as the seminal one for this new calibration approach.

Major Comments:

1. Line 33: Sentence starting "However …" this sentence seems to imply that the technique presented here does not carry the uncertainty of the reference sonde but of course it does. Please revise sentence to eliminate any confusion.
2. Line 40: Sentence starting "What sets this approach …" I disagree that use of a single external reference calibration effectively diminishes the uncertainty stemming from the reference instrument. In fact, use of a single radiosonde for calibration implies that any systematic uncertainties associated with that single sonde will be carried through the entire calibrated time series. This is a distinct weakness in the proposed approach and the reason why an ensemble of sondes is typically used for calibration – the influence of systematic uncertainties due, for example, to sampling issues or sonde/lidar manfunctions will be reduced in the ensemble. Furthermore, use of an ensemble of sondes may permit some characterization of uncertainties in the reference dataset and in the transfer of the calibration to the lidar. Those uncertainties must then be propagated into the uncertainties in the time series. Without an ensemble of sondes, what uncertainty will be attributed to the transfer of the reference calibration? There are at least three terms that must be evaluated to obtain the fully propagated uncertainty of the calibrated time series: 1) the uncertainty in the lidar retrieval, 2) the uncertainty in the radiosonde used for calibration 3) the uncertainty in the transfer of the radiosonde calibration to the lidar time series. How do you evaluate 3) using your current scheme? If an ensemble of sondes is used for calibration, you can use the statistics of these comparisons to estimate the uncertainty in transferring the calibration.
3. Line 104: About the use of 70 deg SZA for solar calibration … given the spectral proximity of the high and low J channels, is any significant difference in the solar spectrum expected? One of the great attractions of this technique would be if one could simply assume that the solar spectral intensity is essentially identical in the spectral windows of HiJ and LoJ. An interrogation of a high resolution radiative transfer model would answer this question. You also could address this question empirically by trying the method at different SZAs to see if there is any difference in the ratios computed. It may be that this ratio is independent of SZA. It may also be that you could use cloudy data just as easily. The question is whether there is a significant difference in the spectral intensity of the source

(whether the sun, clouds, whatever) at the two closely spaced optical channels. That really should be addressed in this seminal work.

4. Figure 2: plot in Fig 2a seems to show a few clear outliers with temperature differences of 10-20 K in the lowest 10 km. Some filtering should be done to eliminate these from the statistics since they are pretty clearly aberrant comparisons. There also are a couple aberrant comparisons in Fig 2b, the background calibration plot, with differences of close to 30 K. If these are not outliers and should be included then some explanation is needed as to why these comparisons are retained.

5. Table 1. Please include standard deviations with all of these values. I suspect that for many (or perhaps most) there will not be a significant difference, even at the 1-sigma level, between the external and solar background methods (particularly after addressing comment 4 above). If so, some of your discussion will need revising. Realize though that even if after addressing this comment there is not a statistically significant improvement in the results, this new technique has the distinct advantage of tracking calibration changes better. You know this, of course. My point is to not sweat it if the statistical results are not better with the background technique. The ability to calibrate once and then track changes is more than enough to warrant use of this new calibration idea.

6. Line 170: "We have shown the solar background calibration method …." You have shown this when used in conjunction with the OEM retrieval method. I suspect most of the community still uses the traditional method a la Behrendt et al. Would it not make your argument more persuasive if you applied your technique also to retrievals using the traditional method? I think you should take the time to perform this work so that you get full credit for establishing the solar background technique as the preferred way to calibrate RR lidar temperature measurements (which you deserve!).

Minor Comments:

1. Line 14: delete first appearance of "system" to avoid redundancy
2. Line 28: change "from" to "from the determination of"
3. Line 30: I would introduce the phrase "Optimal Estimation Method (OEM)" in this sentence so that you can refer to OEM moving forward. For most of us, OEM is more likely to be understood as Original Equipment Manufacturer so best to avoid that confusion up front.
4. Line 63: Claim is made that RALMO meets "stringent OSCAR … requirements … with an uncertainty of less than 1K". Please note that the OSCAR requirements are specified as "goal", "breakthrough" and "threshold". The 1K uncertainty in temperature that is referred to is the "breakthrough" requirement for High Resolution NWP in the Free Troposphere. It would be best to refer explicitly to this "breakthrough" requirement. *BUT* the breakthrough requirement is not just about the level of uncertainty. In order to meet this requirement, it is also necessary to provide this level of uncertainty at 5 km horizontal resolution every 30 minutes for the entire free troposphere. Please clarify if RALMO meets these other parts of the requirement.
5. Line 67: All lidars require calibration whether internal or external. So I would suggest changing this sentence to something like: "Lidar temperature measurements, including those from Raman lidar studied here, require calibration."
6. Line 71: Equation 1. So that the reader does not have to refer back to the earlier paper, I suggest that you reproduce equation 1 from Mahagammulla Gamage et al., 2019. Then

your equations follow more easily where $C_{RR}$ from M-G becomes either $C_{JH}$ or $C_{JL}$ in your terminology. It would also help to maintain use of R from the earlier paper for the ratio shown in your equation 1 as C*. If you prefer to use C* (script R is used for things like aerosol scattering ratio in the literature) then just introduce short text here explaining that your notation is deviation from M-G.

7.  Line 72: reference is made to "RALMO's digital channels". This seems an awkward construction. The lidar constants refer to optical characteristics which are measured, hopefully accurately, by the digital electronics. Suggest changing "RALMO's digital channels" to something like "RALMO's high J and low J rotational Raman channels, respectively."

8.  Line 77: same comment about "digital channels"

9.  Line 78: typo "GRAUN"

10. Line 79: you introduce a new concept that equation 2 has a range dependence which you are ignoring. I think you need to expand on this some. What is the range dependence due to? What is the estimated uncertainty due to this omission?

11. Line 93: as stated in the major comments, I disagree that use of a single sonde for calibration significantly reduces uncertainties. Having a single reference for your time series eliminates jumps in calibration, which is definitely a good thing. The issue is whether your single reference should be obtained from one comparison or from an ensemble. The argument in the major comments is that you should use an ensemble of comparisons to derive the calibration.

12. Line 101: Yes, you can choose any point in the time series for establishing the calibration but it seems awkward to me to call it $t_0$ since this is usually understood to be a start time of an experiment. I would suggest you change this notation to something like t' to avoid this confusion. Or at least I was confused by how you could consider times prior to $t_0$...

13. Line 103: I understand that you are analyzing a historical dataset and thus must avoid the laser return signal. But, our publications have shown Raman nitrogen returns to beyond 55 km (Whiteman et al., 2010 Fig 12) and your rotational Raman signal is likely stronger. So please state what you did to assess that no significant returns were observed from 55 km.

14. Line 109: Regarding the calibration results using the external method. I assume that this work was done as a part of this paper and you are not referring to work performed in an earlier effort. I suggest you make this explicit by, for example, changing the sentence that starts "For the external method..." to "As an illustration of the external calibration method we processed ..."

15. Line 113: Sentence starting "This approach involved ...". Just to point out that this filtering method could be used to choose a set of profiles to investigate whether the ratio in your equation 1 is the same under cloudy conditions as for solar background. You would still use the values above 55 km as before.

16. Line 119: Suggest rewording sentence starting "The t0 ..." to "The date selected for the t' calibration was June 5, 2013.

17. Line 119: You refer to "error values of each external calibration constant". It is not clear what this refers to. Have you done some kind of RMS difference between lidar and sonde profile to determine a difference statistic? Please describe in more detail but I would avoid the use of the term "error" since this implies a reference to an absolute standard and I don't think you can make such a claim for a comparison between sonde and lidar where they are likely sampling different volumes. You can talk about differences or uncertainties but error is likely not a proper term here.

18. Line 122: concerning "using a Licel detection system". What is the significance that the data acquisition electronics were manufactured by Licel? Also, a "detection system" would describe both optical and electrical components I would think. So in reference to Licel I would just call it the data acquisition system.

19. Line 123: Sentence starting "This change …" This is indeed a very nice demonstration of how your background method tracks calibration changes. But I would appreciate some more discussion on why adjusting a coaxial cable would make any changes in the calibration at all. In order to have a dramatic change in the ratio as you show, an effective change in optical/electrical efficiency of one of the channels would have to occur. So please describe briefly how adjusting the coax cable results in such a change. It would be much easier to see this large change in ratio as being due to a change in some optical component, for example, or in a change in a discriminator setting and not just due to changing a cable.

20. Figure 2: typo "GRAUN"

21. Line 128: concerning "much more stable after 2013". Can you offer an explanation for why this is so?

22. Line 130: as noted earlier I suggest introducing the abbreviation OEM when first mentioned.

23. Line 130: "Only digital channel measurements were used …" I suggest that you add somewhere a brief explanation that the Licel electronics offer two simultaneous measurements of the same signal - one using photon counting, the other using voltage measurements determined through an analog to digital converter. Please note that both of these signals are "digital signals" so these signals are usually referred to as photon counting and analog signals.

24. Line 132: "1 instead" is this a typo?

25. Line 134: I would delete the mention of Licel and reword to not use "using" twice in rapid succession. For example, "The measurements for the years 2011 to 2014 were processed first using the externally determined calibration constants and secondly … "

26. Line 135: Awkward construction. Suggest rewording sentence starting "During the data processing employing external calibration coefficients, these coefficients were interpolated …" to "The externally determined calibration coefficients were interpolated …"

27. Line 138: You state "…the calibration constant C*(t0) was computed by averaging over the 5 to 8 km altitude range." I am confused. You are referring to the background calibration technique in this sentence and the value of C* is determined above 55 km, right? It is for the external calibration technique that you use a comparison between 5-8 km. Correct?

28. Line 139: you refer to the "cost associated with the OEM retrieval process". Please add some explanation for what this cost function is.

29. Line 140: Ditto. What are these cost values and why do they imply they should be eliminated. Since you are relying so much on this earlier OEM retrieval work, I think you should add a section describing the OEM details enough so that readers understand the technique along with the cost function and its range of values.

30. Line 140: Instead of "Each dataset" can you name the datasets to avoid confusion? Do you mean externally calibrated and solar background calibrated?

31. Line 142: reference is made to a "measurement response function". Please explain what this is.

32. Line 144: reference is made to "homogenized radiosonde measurements". What are these? Please explain.

33. Line 144: you also refer to "GRUAN certified radiosondes". This is a little misleading it seems to me. GRUAN reprocesses the standard radiosonde products to create a GRUAN certified data product. The radiosondes themselves are the same either way. So the radiosondes are not certified but rather the final processed profiles are certified. You also state that these are "special soundings". I don't think this is right. The difference is that you have sent the standard data products to GRUAN and received re-processed data products. Right? I would make all of this more clear when you explain the difference between homogenized and GRUAN soundings.
34. Line 147: I think instead of "correlating" you mean "corresponding"?
35. Line 147: "data set". Please be consistent throughout the paper either "dataset" or "data set". Personally, I don't care which.
36. Line 151: the statement "...an underestimation of the lidar temperatures in this range" is unclear to me. Suggest : "an underestimation of the lidar-derived temperatures with respect to the radiosonde measured temperatures". Also, I wonder if after including standard deviation statistics in Table 1 this underestimation will be statistically significant...same statement pertains to other assessments of underestimations later in the paper.
37. Line 161: Suggest change "(and unanticipated)" to "(and possibly unanticipated)"

---

## Author Comment (AC1)

**We would like to thank you for your thoughtful review of the manuscript. Your comments have improved the work and its presentation. Your suggestion to expand to the traditional method was particularly valuable, and after some false starts we were able to include how to use this method in the traditional algorithm. We have carefully considered and addressed each of your points, and we believe that the revisions have significantly improved the clarity and robustness of the paper.**

Major Comments:

1. Line 33: Sentence starting "However …" this sentence seems to imply that the technique presented here does not carry the uncertainty of the reference sonde but of course it does. Please revise sentence to eliminate any confusion.

   We agree and have revised the sentence accordingly. Please refer to line 36 in the revised manuscript with track changes.

2. Line 40: Sentence starting "What sets this approach …" I disagree that use of a single external reference calibration effectively diminishes the uncertainty stemming from the reference instrument. In fact, use of a single radiosonde for calibration implies that any systematic uncertainties associated with that single sonde will be carried through the entire calibrated time series. This is a distinct weakness in the proposed approach and the reason why an ensemble of sondes is typically used for calibration – the influence of systematic uncertainties due, for example, to sampling issues or sonde/lidar malfunctions will be reduced in the ensemble. Furthermore, use of an ensemble of sondes may permit some characterization of uncertainties in the reference dataset and in the transfer of the calibration to the lidar. Those uncertainties must then be propagated into the uncertainties in the time series. Without an ensemble of sondes, what uncertainty will be attributed to the transfer of the reference calibration? There are at least three terms that must be evaluated to obtain the fully propagated uncertainty of the calibrated time series: 1) the uncertainty in the lidar retrieval, 2) the uncertainty in the radiosonde used for calibration 3) the uncertainty in the transfer of the radiosonde calibration to the lidar time series. How do you evaluate 3) using your current scheme? If an ensemble of sondes is used for calibration, you can use the statistics of these comparisons to estimate the uncertainty in transferring the calibration.

   We agree that employing an ensemble of external calibrations (radiosondes) is the better approach; thus, we have redone the validation using an ensemble rather than a single calibration point at time t0. The equations have been adjusted accordingly, and the figures of the calibration time series and the temperature difference plot have been updated to reflect the new results derived from the ensemble approach. This ensemble-based method improves accuracy, which can now be characterized by the RMS. Please refer to lines 176-180 in the revised manuscript with track changes.

3. Line 104: About the use of 70 deg SZA for solar calibration ... given the spectral proximity of the high and low J channels, is any significant difference in the solar spectrum expected? One of the great attractions of this technique would be if one could simply assume that the solar spectral intensity is essentially identical in the spectral windows of HiJ and LoJ. An interrogation of a high resolution radiative transfer model would answer this question. You also could address this question empirically by trying the method at different SZAs to see if there is any difference in the ratios computed. It may be that this ratio is independent of SZA. It may also be that you could use cloudy data just as easily. The question is whether there is a significant difference in the spectral intensity of the source (whether the sun, clouds, whatever) at the two closely spaced optical channels. That really should be addressed in this seminal work.

   We tested our method across various solar zenith angles (70°, 60°, and 50°), and the results indicated an average variation of 0.2% in the calibration constant, corresponding to a temperature change of approximately 0.2 K. This information has been included in the revised manuscript. Please refer to line 186 in the revised manuscript with track changes.

4. Figure 2: plot in Fig 2a seems to show a few clear outliers with temperature differences of 10-20 K in the lowest 10 km. Some filtering should be done to eliminate these from the statistics since they are pretty clearly aberrant comparisons. There also are a couple aberrant comparisons in Fig 2b, the background calibration plot, with differences of close to 30 K. If these are not outliers and should be included then some explanation is needed as to why these comparisons are retained.

   Additional filtering, which involved removing profiles exhibiting unphysical features in the raw signal, combined with the switch to an ensemble approach, has effectively eliminated many of the aberrant comparisons at lower altitudes. Please refer to line 245 in the revised manuscript with track changes.

5. Table 1. Please include standard deviations with all of these values. I suspect that for many (or perhaps most) there will not be a significant difference, even at the 1-sigma level, between the external and solar background methods (particularly after addressing comment 4 above). If so, some of your discussion will need revising. Realize though that even if after addressing this comment there is not a statistically significant improvement in the results, this new technique has the distinct advantage of tracking calibration changes better. You know this, of course. My point is to not sweat it if the statistical results are not better with the background technique. The ability to calibrate once and then track changes is more than enough to warrant use of this new calibration idea.

   The table has been updated with the new statistics, including the standard deviation obtained by employing an ensemble of radiosondes. The discussion has also been

revised to reflect these updated statistics. Please refer to table 1 and lines 276-292 in the revised manuscript with track changes.

6. Line 170: "We have shown the solar background calibration method ...." You have shown this when used in conjunction with the OEM retrieval method. I suspect most of the community still uses the traditional method a la Behrendt et al. Would it not make your argument more persuasive if you applied your technique also to retrievals using the traditional method? I think you should take the time to perform this work so that you get full credit for establishing the solar background technique as the preferred way to calibrate RR lidar temperature measurements (which you deserve!).

Thank you very much for your input. We have developed the theoretical framework and incorporated it into Subsection 2.6. Our tests on our Lidar system demonstrate that the relationship between Q and T holds as long as the spectral characteristics of the receiver remain constant. We do not include the results from our tests in the manuscript because we don't want the focus to shift too much to the traditional method. Nevertheless, our findings illustrate a clear path for implementing this method within traditional retrieval processes. Additionally, given the range of conversion relationships between Q and T, we recognize that a comprehensive exploration of these methods would extend beyond the scope of our current study.

Minor Comments:

1. Line 14: delete first appearance of "system" to avoid redundancy

   Fixed

2. Line 28: change "from" to "from the determination of"

   Fixed

3. Line 30: I would introduce the phrase "Optimal Estimation Method (OEM)" in this sentence so that you can refer to OEM moving forward. For most of us, OEM is more likely to be understood as Original Equipment Manufacturer so best to avoid that confusion up front.

   Fixed

4. Line 63: Claim is made that RALMO meets "stringent OSCAR ... requirements ... with an uncertainty of less than 1K". Please note that the OSCAR requirements are specified as "goal", "breakthrough" and "threshold". The 1K uncertainty in

temperature that is referred to is the "breakthrough" requirement for High Resolution NWP in the Free Troposphere. It would be best to refer explicitly to this "breakthrough" requirement. BUT the breakthrough requirement is not just about the level of uncertainty. In order to meet this requirement, it is also necessary to provide this level of uncertainty at 5 km horizontal resolution every 30 minutes for the entire free troposphere. Please clarify if RALMO meets these other parts of the requirement.

We agree and have amended the text as follows: "RALMO meets the OSCAR requirements at breakthrough level for High Resolution NWP in the Free Troposphere in terms of measurement uncertainty and observing cycle." Please refer to line 81 in the revised manuscript with track changes.

5. Line 67: All lidars require calibration whether internal or external. So, I would suggest changing this sentence to something like: "Lidar temperature measurements, including those from Raman lidar studied here, require calibration."

Fixed

6. Line 71: Equation 1. So that the reader does not have to refer back to the earlier paper, I suggest that you reproduce equation 1 from Mahagammulla Gamage et al., 2019. Then your equations follow more easily where CRR from M-G becomes either CJH or CJL in your terminology. It would also help to maintain use of R from the earlier paper for the ratio shown in your equation 1 as C*. If you prefer to use C* (script R is used for things like aerosol scattering ratio in the literature) then just introduce short text here explaining that your notation is deviation from M-G.

Fixed. We also added a few lines explaining the adaptation of the variable C*. Please refer to line 145 in the revised manuscript with track changes.

7. Line 72: reference is made to "RALMO's digital channels". This seems an awkward construction. The lidar constants refer to optical characteristics which are measured, hopefully accurately, by the digital electronics. Suggest changing "RALMO's digital channels" to something like "RALMO's high J and low J rotational Raman channels, respectively."

Fixed

8. Line 77: same comment about "digital channels"

Fixed

9. Line 78: typo "GRAUN"

Fixed

10. Line 79: you introduce a new concept that equation 2 has a range dependence which you are ignoring. I think you need to expand on this some. What is the range dependence due to? What is the estimated uncertainty due to this omission?

We don't ignore the range dependence, we omit its notation for improved readability in equation 5. We amended the text as follows: "Equation 5 can in principle be evaluated at any altitude and we have omitted the range dependence for improved readability. We calculated the calibration constants by averaging over the 5 to 8 km altitude range to reduce the random error and to avoid regions of the profiles where the signals could be saturated." Please refer to line 155 in the revised manuscript with track changes.

11. Line 93: as stated in the major comments, I disagree that use of a single sonde for calibration significantly reduces uncertainties. Having a single reference for your time series eliminates jumps in calibration, which is definitely a good thing. The issue is whether your single reference should be obtained from one comparison or from an ensemble. The argument in the major comments is that you should use an ensemble of comparisons to derive the calibration.

We agree and are now using an ensemble. The equations have been updated accordingly, please refer to equation 7 in the revised manuscript with track changes.

12. Line 101: Yes, you can choose any point in the time series for establishing the calibration, but it seems awkward to me to call it t0 since this is usually understood to be a start time of an experiment. I would suggest you change this notation to something like t' to avoid this confusion. Or at least I was confused by how you could consider times prior to t0.

This is obsolete in the revised manuscript after switching to an ensemble.

13. Line 103: I understand that you are analyzing a historical dataset and thus must avoid the laser return signal. But, our publications have shown Raman nitrogen returns to beyond 55 km (Whiteman et al., 2010 Fig 12) and your rotational Raman signal is likely stronger. So please state what you did to assess that no significant returns were observed from 55 km.

This does not present an issue during the daytime, as with strong solar background any return above 55 km is negligible! For nighttime, we extrapolated our N2 raw signal to estimate the expected return at 55 km and compared it to the actual background value at this altitude. Our findings indicate that the background is 2000 times larger than the anticipated signal return.

14. Line 109: Regarding the calibration results using the external method. I assume that this work was done as a part of this paper, and you are not referring to work performed in an earlier effort. I suggest you make this explicit by, for example,

changing the sentence that starts "For the external method…" to "As an illustration of the external calibration method we processed …"

Fixed.

15. Line 113: Sentence starting "This approach involved …". Just to point out that this filtering method could be used to choose a set of profiles to investigate whether the ratio in your equation 1 is the same under cloudy conditions as for solar background. You would still use the values above 55 km as before.

Thanks for the suggestion. We are keeping this for future investigations.

16. Line 119: Suggest rewording sentence starting "The t0 …" to "The date selected for the t' calibration was June 5, 2013.

This is obsolete in the revised manuscript after switching to an ensemble.

17. Line 119: You refer to "error values of each external calibration constant". It is not clear what this refers to. Have you done some kind of RMS difference between lidar and sonde profile to determine a difference statistic? Please describe in more detail but I would avoid the use of the term "error" since this implies a reference to an absolute standard and I don't think you can make such a claim for a comparison between sonde and lidar where they are likely sampling different volumes. You can talk about differences or uncertainties but error is likely not a proper term here.

This whole phrase became obsolete after switching to the ensemble method. We are not using the uncertainties of the external calibration constants in the calculations.

18. Line 122: concerning "using a Licel detection system". What is the significance that the data acquisition electronics were manufactured by Licel? Also, a "detection system" would describe both optical and electrical components I would think. So in reference to Licel I would just call it the data acquisition system.

We agree, and we have replaced detection system by the acquisition system. We have added a few sentences explaining that our dataset is split into data acquired with licel and with fastcom and here we are working only with licel. See comment 23. Also please refer to lines 63-70 in the revised manuscript with track changes.

19. Line 123: Sentence starting "This change …" This is indeed a very nice demonstration of how your background method tracks calibration changes. But I would appreciate some more discussion on why adjusting a coaxial cable would make any changes in the calibration at all. In order to have a dramatic change in the ratio as you show, an effective change in optical/electrical efficiency of one of the channels would have to occur. So please describe briefly how adjusting the coax cable results in such a change.

It would be much easier to see this large change in ratio as being due to a change in some optical component, for example, or in a change in a discriminator setting and not just due to changing a cable.

The replacement of JL's coax cable connecting the PMT with the Licel recorder is the only detail we can find in the logbook of the lidar. And since the event is already more than ten years ago, the involved scientists and technicians are either not available anymore or do not remember any further details. We looked at the solar background from the internal calibrations and can see an increase in counts in JL pointing to a better sensitivity following the intervention. This better sensitivity is consistent with a drop in the calibration constant. We amended the text referring to the logbook leaving it open between the lines, that it might be incomplete. Since the scope of the work is not on the system, we feel this should be sufficient. Please refer to line 223 in the revised manuscript with track changes.

20. Figure 2: typo "GRAUN"

    Fixed

21. Line 128: concerning "much more stable after 2013". Can you offer an explanation for why this is so?

    The intervention in 2013 led to a higher sensitivity in the JL channel (see comment above). This could explain the fact that the calibration is less noisy after 2013. We amended the text as follows: "... less noisy after the intervention on the JL channel in 2013." Please refer to line 229 in the revised manuscript with track changes.

22. Line 130: as noted earlier I suggest introducing the abbreviation OEM when first mentioned.

    Fixed

23. Line 130: "Only digital channel measurements were used ..." I suggest that you add somewhere a brief explanation that the Licel electronics offer two simultaneous measurements of the same signal - one using photon counting, the other using voltage measurements determined through an analog to digital converter. Please note that both of these signals are "digital signals" so these signals are usually referred to as photon counting and analog signals.

    We added a small paragraph under RALMO explaining the two modes of detection. Please refer to line 63 in the revised manuscript with track changes.

24. Line 132: "1 instead" is this a typo?

This refers to equation 1 (now equation 4). We added the word 'equation' to avoid confusion. Please refer to line 234 in the revised manuscript with track changes.

25. Line 134: I would delete the mention of Licel and reword to not use "using" twice in rapid succession. For example, "The measurements for the years 2011 to 2014 were processed first using the externally determined calibration constants and secondly …"

Fixed

26. Line 135: Awkward construction. Suggest rewording sentence starting "During the data processing employing external calibration coefficients, these coefficients were interpolated …" to "The externally determined calibration coefficients were interpolated …"

Fixed

27. Line 138: You state "…the calibration constant C*(t0) was computed by averaging over the 5 to 8 km altitude range." I am confused. You are referring to the background calibration technique in this sentence and the value of C* is determined above 55 km, right? It is for the external calibration technique that you use a comparison between 5-8 km. Correct?

Yes C*(t0) now $\overline{C^{\cdot}(t)}$ represents the average of the ensemble of external calibrations. Each of these points is computed using equation 5 averaged over the 5-8 km altitude range.

28. Line 139: you refer to the "cost associated with the OEM retrieval process". Please add some explanation for what this cost function is.

Explanation added under the new OEM subsection. Please refer to equation 2 in the revised manuscript with track changes.

29. Line 140: Ditto. What are these cost values and why do they imply they should be eliminated. Since you are relying so much on this earlier OEM retrieval work, I think you should add a section describing the OEM details enough so that readers understand the technique along with the cost function and its range of values.

We agree. We have added a sentence explaining that these cost values refer to overfitting and underfitting and therefore are used to discard certain retrievals. Please refer to line 244 in the revised manuscript with track changes.

30. Line 140: Instead of "Each dataset" can you name the datasets to avoid confusion? Do you mean externally calibrated and solar background calibrated?

Yes, here we refer to the externally calibrated and solar background calibrated data sets. Fixed. Please refer to line 243 in the revised manuscript with track changes.

31. Line 142: reference is made to a "measurement response function". Please explain what this is.

    Fixed.

32. Line 144: reference is made to "homogenized radiosonde measurements". What are these? Please explain.

    This is addressed under the new subsection radiosondes. Please refer to lines 95-105 in the revised manuscript with track changes.

33. Line 144: you also refer to "GRUAN certified radiosondes". This is a little misleading it seems to me. GRUAN reprocesses the standard radiosonde products to create a GRUAN certified data product. The radiosondes themselves are the same either way. So the radiosondes are not certified but rather the final processed profiles are certified. You also state that these are "special soundings". I don't think this is right. The difference is that you have sent the standard data products to GRUAN and received re-processed data products. Right? I would make all of this more clear when you explain the difference between homogenized and GRUAN soundings.

    This is addressed under the new subsection radiosondes. Please refer to lines 85-94 in the revised manuscript with track changes.

34. Line 147: I think instead of "correlating" you mean "corresponding"?

    Fixed.

35. Line 147: "data set". Please be consistent throughout the paper either "dataset" or "data set". Personally, I don't care which.

    Fixed.

36. Line 151: the statement "...an underestimation of the lidar temperatures in this range" is unclear to me. Suggest : "an underestimation of the lidar-derived temperatures with respect to the radiosonde measured temperatures". Also, I wonder if after including standard deviation statistics in Table 1 this underestimation will be statistically significant...same statement pertains to other assessments of underestimations later in the paper.

    Fixed.
37. Line 161: Suggest change "(and unanticipated)" to "(and possibly unanticipated)"
    Fixed

---

## Author Comment (AC2)

**We appreciate your detailed and insightful comments, which have helped improve the clarity and precision of our work. Each point has been thoughtfully addressed, and we believe these revisions have enhanced the manuscript. Thank you for your contribution to this work.**

1) There is no general description of the RALMO lidar. The paper should al least have a simple but adequate schematic of which channels are used and how these channels come about. An accompanying table listing the instrument specifications should also be added.

We include references to detailed descriptions of RALMO, further, our method is not specific to the lidar. Hence, we do not agree that a schematic is needed here.

2) The system constants in Eq. 1 require a bit more introduction. The lidar equation for PRR temperature should be introduced first.

We agree. We added the lidar equation and explained the relevant variables. Please refer to equation 3 and lines137-141 in the revised manuscript with track changes.

3) 'GRUAN', not 'GRAUN'

Fixed

4) In contrast to the ack of an overall system overview, the Licel (digitisation system) is mentioned a couple of times, without explaining what it is. Is it the Analog/photon counting combination that is referred to or merely the digitised signals?

We added a paragraph explaining the Licel data acquisition system.  Please refer to line 63 in the revised manuscript with track changes.

5) A number of references to in-depth descriptions of the Raman lidar technique for water vapour and temperature lidar, including error analysis are missing. E.g.
- David N. Whiteman, "Examination of the traditional Raman lidar technique. I. Evaluating the temperature-dependent lidar equations," Appl. Opt. 42, 2571-2592 (2003).
- Leblanc, T., Sica, R. J., van Gijsel, J. A. E., Haefele, A., Payen, G., and Liberti, G.: Proposed standardized definitions for vertical resolution and uncertainty in the NDACC lidar ozone and temperature algorithms – Part 3: Temperature uncertainty budget, Atmos. Meas. Tech., 9, 4079–4101, https://doi.org/10.5194/amt-9-4079-2016, 2016.

We agree and have included Whiteman's reference under the RALMO section (line 74). However, we consider the Leblanc paper to be unrelated to our study, as it focuses on Rayleigh temperature measurements, which do not align with the scope of our work.

6) I recommend restructiring the manuscript to first clearly describe the methods used to retrieve the temperature profiles.

- o In the results section an optimal estimation method appears out of the blue. The term OEM is not explained.

  We agree. We added a subsection explaining the basics of OEM along with the relevant equations. Please refer to subsection 2.3 in the revised manuscript with track changes.

- o Also a description of the GRUAN sonde products is needed. Was the uncertainty information in the GRUAN profiles used?

  We agree and have now added a new subsection briefly explaining about the radiosondes used. We also referenced the papers for the RS92 and RS41 GDP (GRUAN data product) for more detailed information. We do not use the uncertainty of the GDPs. Please refer to subsection 2.2 in the revised manuscript with track changes.

7) Please add some clarifying labels to Figure 1 to guide the reader. E.g. where is t0? What happens sometime in 2012 (make reference to the text) and also just before 2013?

As per Whiteman's suggestion, we have adopted the use of an ensemble of radiosondes rather than relying on a single radiosonde calibration at t0. Please refer to major comment #2 from Whiteman for further details.  Hence, this is obsolete after switching to the ensemble approach. We have also explained the drop observed in 2012.  Please refer to line 223 in the revised manuscript with track changes.

8) I believe that the continuous background method can be used to monitor sudden changes in instrument behaviour by following the value of the calibration constant. However, an external source is always needed for an absolute calibration. Therefore, the accuracy is always limited by the accuray and uncertainty of the external reference and, in this case the uncertainties of the lidar. This needs to be further elaborated.

This is correct. Now with ensemble the accuracy is improved and can be characterized with the RMS. While the absolute value depends entirely on the external reference, the trend is to the greatest extent independent of the external reference. Furthermore, we are very transparent that the background calibration still requires an absolute reference and that it's the trend that becomes essentially independent from the external reference. Please refer to lines 36 and 53 in the revised manuscript with track changes.

---

## Author Comment (AC3)

1. In addition to the changes requested by the two reviewers, I would request a line or two be added somewhere to acknowledge that the stability of the RR temperatures can also depend on the laser stability. If the laser line drifts around by a few picometres due to temperature changes in the laser, then you could see a change in the retrieved RR temperatures up to a Kelvin. The optimal solution on the transmitter side is external seeding and a laser pulse spectrometer to measure the offset between the seed laser and the output of the power laser.

Thank you for your comment. Our laser is unseeded, with a linewidth of 1 cm$^{-1}$ (30 GHz) as specified by Litron (communication on 25 April 2018). The temperature dependency of the laser line center is 0.004 nm/°C at 1064 nm, which translates to approximately 1.3 pm/°C at 355 nm (around 3 GHz/°C or 0.1 cm$^{-1}$/°C). Considering variations of ±3°C in the cooling system, this would result in a displacement of the laser frequency by ~0.3 cm$^{-1}$. However, the PRR polychromator linewidth is estimated to be 20-25 cm$^{-1}$ (600-750 GHz), based on the RALMO user manual (EPFL) and Martucci et al. (2021). This corresponds to the spectral width of the Stokes and Anti-Stokes spectra diffracted by the grating filter onto each optic fiber in the polychromator. Given that the laser is stabilized within a temperature range of ±2°C by the water-cooling system, even a change of 3°C would only cause a relative displacement of 1/80th of the PRR polychromator spatial bandwidth. Thus, it would not affect the polychromator's ability to select the central main Stokes and Anti-Stokes lines (Tables 1-2, Martucci et al. 2021).

---

## Referee Report (RR1)

**Solar Background Radiation Temperature Calibration of a Pure Rotational Raman Lidar, Jayaweera et al. Revised – a Review**

David N. Whiteman, Howard University

**General Comments**

The simplicity of this technique is very compelling and should become a standard part of calibration for rotational Raman lidar temperature measurements. The authors have shown that the technique provides results consistent with and sometimes better than the traditional calibration technique and permits tracking changes in system configuration that would likely be missed if only episodic external calibrations are performed. I believe the paper should be published after some fairly minor revisions.

I do have some significant concerns with the manuscript as it currently stands which I detail in the major comments. Perhaps the most important of the major comments is the one concerning the determination of $r_{solar}$. The authors really should take advantage of the demonstrated insensitivity of the $r_{solar}$ ratio to changes in SZA. Two closely spaced wavelength intervals illuminated by broadband light in the absence of significant absorbers should be essentially constant as shown in a limited way by the authors. This suggests that the ratio $r_{solar}$ could be determined under a broader range of conditions than studied by the authors. This is a great strength of the technique and should be emphasized as such.

In the minor comments provided, I have taken the liberty to offer rather detailed changes with the hope that this paper becomes a standard reference in the literature.

**Major Comments**

1. Line 138/Eq 3:
   1. I believe that the equation as written is incorrect. The number density n(z) as used is the volume mixing ratio of both nitrogen and oxygen. But the scattering cross section of these molecules is different and the current equation does not capture this. I suggest following a notation similar to what is used in either the numerator or denominator of eq 4 in Adam et al., "Notes on Temperature-Dependent Lidar Equations" JTECH, 2009 Vol 26, 1021-1039, (or the notation you use in your eq 8 below) where the number density of either $N_2$ or $O_2$ is now inside of the larger summation over the two different molecules

$$F_L(T) = \frac{\sum\limits_{n=N_2,O_2} \eta_n \sum\limits_i \left[\frac{d\sigma(\lambda_{X,i}, T, \pi)}{d\Omega}\right]_n \xi(\lambda_{X,i})}{\sum\limits_{n=N_2,O_2} \eta_n \left[\frac{d\sigma_t(\lambda_X, \pi)}{d\Omega}\right]_n \xi(\lambda_X)}, \quad (4)$$

   2. Also, I had some trouble reconciling the notation in equations 3 and 4. I think my trouble came from what appears to be mixed usage of the notation "RR". In equation 3,

the notation RR usually stands for either the entire signal of $J_H$ or $J_L$ (as in the cases of $N_{RR}$, $C_{RR}$, $B_{RR}$). However, the use of RR in $tau_{RR}$ refers to just a single line and this is confusing. I suggest 1) dropping the RR subscript from tau and, in general, changing RR to Jx with the explanation that x can be either H or L referring to either the high or low RR channel. With this change in notation, equation 4 follows more naturally.

2. Line 187, sentence starting "We tested ..."
   1. A variation of 0.2% is surely within the uncertainty of the technique. If so, this result implies that the solar background value is independent of SZA and the authors should so state. Such a result would make the technique more robust and easier to implement. Please add a sentence following the one written something like "This result suggests that $r_{solar}$ is independent of solar zenith angle as would be expected for the two closely spaced wavelength intervals.
3. Lines 284-293: Please be careful with this discussion. Four of the six comparisons presented do not differ beyond the uncertainty bars indicated (1-sigma?). Thus it would seem justified to state that the background and external methods yield similar results for those 4 metrics whereas for the other 2 metrics (mean bias 4-8 and mean IQR 1-8) the background method shows better agreement at the 1-sigma level of significance.

**Minor Comments**

1. Lines 5, 45: Suggest "rotational Raman temperature lidar"
2. Line 53: "reduced accuracy" → "reduced precision"
   1. the term accuracy refers to a deviation from truth whereas precision refers to the spread of a set of measurements. Use of a larger ensemble of radiosondes should reduce the uncertainty in the mean value but, unfortunately, does not necessarily guarantee that we are closer to the truth.
3. Line 64: Suggest "using Licel GmbH transient recorders, which enable ..."
4. Line 112: "rotational temperature" → "rotational Raman temperature"
5. Line 127: please italicize "*a priori*"
6. Line 136: Why use PRR here? Why not NRR? Then line 139 becomes simply "where NRR is the true backscattered signal ..."
7. Line 142: Suggest "Following the methodology of MG (2019) we can define ..."
8. Line 158: "random error" → "random uncertainty"
   1. same comment as in 2 above
9. Lines 139-140: "geometrical overlap" → "overlap"
   1. O(z) is the entire overlap function which consists of geometrical and optical components.
10. Line 141: I believe that "attenuated" should be dropped. The attenuation is being accounted for in the equation elsewhere.
11. Line 158: "random error" → "random uncertainty"
12. Lines 162-163: Suggest "Depending on the external instrument's operating schedule..."
13. Line 169: I was not clear on what "This approach" referred to. Suggest "The approach that we present here mirrors ..."
14. Line 170: "mixing calibration" → "mixing ratio calibration"
15. Line 171: "this approach" → "the approach here"
16. Line 171: "only one reference radiosonde measurement" → "a single calibration based on an ensemble of radiosondes"

17. Line 173: Suggest "drifts"→"calibration changes" to distinguish from the recently used different context of drift.
18. Line 173: Instead of "The relative calibration ..." as written, I suggest starting a new paragraph with something like: "We now define the relative calibration time series as follows". I believe that such a sentence will better prepare the reader for equation 6
19. Line 177: C* was earlier called just the calibration constant. I also think "derive" is the wrong word to use here. Instead I suggest: " We can now use $r_{solar}$ to calculate the time series of the calibration constant C*, the function ..."
20. Line 210: "has been launching" → "had been launching"
21. Line 218: sentence starting "We employed ...". This has already been stated, right? I suggest deleting this sentence.
22. Line 222: "measurement"→"measurements"
23. Line 224: "details"→"detail"
24. Line 225: suggest "low J's photomultiplier with" → "the low-J channel photomultiplier to"
25. Line 231: suggest "within a mean difference of less" → "better"
26. Line 297: "which decreases" → "which can decrease"

---

## Author Response (AR3)

**Solar Background Radiation Temperature Calibration of a Pure Rotational Raman Lidar, Jayaweera et al.**

**08.01.2025**

**Response to Reviewer 1**

We would like to express our sincere gratitude for your thoughtful comments and insightful suggestions. We have addressed each of your points and revised the manuscript accordingly. Your feedback has been invaluable in refining the presentation and clarity of our findings.

Major Comments

1) I believe that the equation as written is incorrect. The number density n(z) as used is the volume mixing ratio of both nitrogen and oxygen. But the scattering cross section of these molecules is different, and the current equation does not capture this. I suggest following a notation similar to what is used in either the numerator or denominator of eq 4 in Adam et al., "Notes on Temperature-Dependent Lidar Equations" JTECH, 2009 Vol 26, 1021-1039, (or the notation you use in your eq 8 below) where the number density of either N2 or O2 is now inside of the larger summation over the two different molecules.

   We believe the lidar equation is correct and have updated the equation to include the volume mixing ratio term within the larger summation. Additionally, we have adjusted equation 8 (eq for Q(t)) to align with this modification in the lidar equation.

   Please refer to equations 3 and 8 in the revised manuscript which tracks changes.

2) Also, I had some trouble reconciling the notation in equations 3 and 4. I think my trouble came from what appears to be mixed usage of the notation "RR". In equation 3, the notation RR usually stands for either the entire signal of JH or JL (as in the cases of NRR, CRR, BRR). However, the use of RR in tauRR refers to just a single line and this is confusing. I suggest 1) dropping the RR subscript from tau and, in general, changing RR to Jx with the explanation that x can be either H or L referring to either the high or low RR channel. With this change in notation, equation 4 follows more naturally.

   We agree with the suggestion and all instances of RR in the Lidar equation and subsequent equations have been replaced with JX, with X explicitly defined as either H or L.

   Please refer to lines 130-137 in the revised manuscript with tracked changes.

3) Line 187, sentence starting "We tested …"A variation of 0.2% is surely within the uncertainty of the technique. If so, this result implies that the solar background value is independent of SZA and the authors should so state. Such a result would make the technique more robust and easier to implement. Please add a sentence following the one

written something like "This result suggests that rsolar is independent of solar zenith angle as would be expected for the two closely spaced wavelength intervals.

We agree and have added explanatory sentences under subsection 2.5 and the conclusion section to emphasize the method's insensitivity to the variations in solar zenith angles.

Please refer to lines 180-181 and 272-274 in the revised manuscript with tracked c hanges.

4) Lines 284-293: Please be careful with this discussion. Four of the six comparisons presented do not differ beyond the uncertainty bars indicated (1-sigma?). Thus it would seem justified to state that the background and external methods yield similar results for those 4 metrics whereas for the other 2 metrics (mean bias 4-8 and mean IQR 1-8) the background method shows better agreement at the 1-sigma level of significance.

We agree with the recommendation and have revised the final paragraph of the results and discussion section.

Please refer to lines 250-265 in the revised manuscript with tracked changes.

Minor Comments

All minor comments have been addressed as per the suggestions provided.

Thanks again for your careful review of the manuscript.